# Acute ACAT1/SOAT1 Blockade Increases MAM Cholesterol and Strengthens ER-Mitochondria Connectivity

**DOI:** 10.3390/ijms24065525

**Published:** 2023-03-14

**Authors:** Taylor C. Harned, Radu V. Stan, Ze Cao, Rajarshi Chakrabarti, Henry N. Higgs, Catherine C. Y. Chang, Ta Yuan Chang

**Affiliations:** 1Department of Biochemistry and Cell Biology, Geisel School of Medicine, Dartmouth College, Hanover, NH 03755, USA; taylor.c.harned.gr@dartmouth.edu (T.C.H.); radu.v.stan@dartmouth.edu (R.V.S.); henry.n.higgs@dartmouth.edu (H.N.H.); 2Chinese Academy of Sciences, Beijing 100045, China; zecao@sioc.ac.cn; 3Department of Pathology, Anatomy and Cell Biology, Thomas Jefferson University, Philadelphia, PA 19107, USA; rajarshi.chakrabarti@jefferson.edu

**Keywords:** cholesterol, ACAT1, SOAT1, ACAT inhibitors, endoplasmic reticulum, mitochondria-associated membrane (MAM), Alzheimer’s disease, K-604, F12511, lipid metabolism

## Abstract

Cholesterol is a key component of all mammalian cell membranes. Disruptions in cholesterol metabolism have been observed in the context of various diseases, including neurodegenerative disorders such as Alzheimer’s disease (AD). The genetic and pharmacological blockade of acyl-CoA:cholesterol acyltransferase 1/sterol O-acyltransferase 1 (ACAT1/SOAT1), a cholesterol storage enzyme found on the endoplasmic reticulum (ER) and enriched at the mitochondria-associated ER membrane (MAM), has been shown to reduce amyloid pathology and rescue cognitive deficits in mouse models of AD. Additionally, blocking ACAT1/SOAT1 activity stimulates autophagy and lysosomal biogenesis; however, the exact molecular connection between the ACAT1/SOAT1 blockade and these observed benefits remain unknown. Here, using biochemical fractionation techniques, we observe cholesterol accumulation at the MAM which leads to ACAT1/SOAT1 enrichment in this domain. MAM proteomics data suggests that ACAT1/SOAT1 inhibition strengthens the ER-mitochondria connection. Confocal and electron microscopy confirms that ACAT1/SOAT1 inhibition increases the number of ER-mitochondria contact sites and strengthens this connection by shortening the distance between these two organelles. This work demonstrates how directly manipulating local cholesterol levels at the MAM can alter inter-organellar contact sites and suggests that cholesterol buildup at the MAM is the impetus behind the therapeutic benefits of ACAT1/SOAT1 inhibition.

## 1. Introduction

Cholesterol is a lipid molecule present in all mammalian cell membranes. Consisting of a rigid sterol-ring backbone, hydrocarbon tail and a hydroxyl group on the 3′ carbon, cholesterol interacts with phospholipids and sphingolipids within the lipid bilayer [1,2]. This interaction introduces structure to the membrane which influences membrane rigidity, fluidity, thickness and integrity [3,4,5]. Higher concentrations of cholesterol and sphingolipids can lead to the self-assembly of liquid-ordered domains, also known as lipid rafts or raft-like domains [6]. These ordered domains were first biochemically isolated as detergent-resistant membrane fractions [7], and have since been observed and extensively studied on the cholesterol and sphingolipid-rich plasma membrane(PM) [8,9,10,11]. The unique physical properties of these domains create lateral heterogeneity by recruiting specific proteins and lipids [6,12,13]. Raft-driven compartmentalization of cellular components is crucial for maintaining cell health through their involvement in membrane trafficking and signal transduction [14]. Whether or not ordered domains can exist on relatively cholesterol-poor intracellular membranes has long-been a question. Recently, ordered domains on the endoplasmic reticulum (ER) were observed at the point of contact between the ER and the mitochondria [15]: a well-studied focal-point of many important homeostatic processes.

The connection between the ER and mitochondria was first observed as early as the 1950s using electron microscopy [16]. This region of the ER, also known as the mitochondria-associated ER membrane (MAM), was first-described to be enriched with lipid-metabolic properties [17], and has since been shown to play host to a range of other important metabolic processes such as calcium homeostasis, mitochondrial division, and autophagosome biogenesis, just to name a few [18]. This inter-organellar contact site is characterized by closely apposed membranes, 10–80 nm apart, which are stabilized by peptide tethers [19,20]. Additionally, compared to the rest of the ER, the MAM is rich in cholesterol and sphingolipids such as ceramide, glucosylceramide and sphingomyelin: all raft-forming lipids [21]. Recent work has demonstrated that ordered domains do, in fact, form at the ER-mitochondria contact site [15], and it has been suggested that modulating cholesterol levels at this domain can influence connectivity [22,23].

Because of its importance in governing membrane function, cholesterol levels within the cell are tightly regulated. The bulk of this cholesterol sensing and regulation occurs on the relatively cholesterol-poor ER. Cholesterol synthesis occurs over a series of reactions catalyzed by enzymes found on the ER [24], and the well-characterized cholesterol sensing mechanisms that regulate the expression of many of those synthesis enzymes are also present at the ER [25]. The removal of cholesterol from the cell is largely facilitated by a family of ATP-binding cassette (ABC) proteins, most prominently, ABCA1 [26]. ABCA1 expression is regulated by cholesterol metabolites known as oxysterols through their activation of nuclear liver X receptors (LXR) [27]. Another mechanism by which the cell can regulate cholesterol in the membrane is through the activity of acyl-CoA:cholesterol acyltransferase (ACAT); also called sterol O-acyltransferase (SOAT). ACAT catalyzes the esterification of membrane-associated cholesterol with an activated fatty acid to produce membrane-incompatible cholesteryl ester (Appendix A). Cholesteryl esters coalesce in cytosolic lipid droplets, effectively removing cholesterol from the membrane [28]. In mammals, ACAT has two isoforms, ACAT1 and ACAT2. ACAT1, gene name *SOAT1*, is widely expressed throughout the body whereas ACAT2 (gene *SOAT2*) expression is limited to the liver and intestinal enterocytes [28]. ACAT activity is primarily regulated post-transcriptionally by substrate availability, and both isoforms are allosterically regulated by cholesterol and oxysterol substrates; however, substrate levels do not affect the protein level [29,30]. ACAT1/SOAT1 is a 9-transmembrane domain protein that resides on the ER as a homotetramer [31,32,33,34,35]. ACAT1/SOAT1 has also been shown to be enriched at the metabolically important subdomain of the ER known as MAM [36,37].

Due to its critical influence on cell function, it is no surprise that disruptions in cholesterol metabolism can lead to disease. One such example is Alzheimer’s disease (AD). AD is estimated to effect 6.5 million people in the United States, a number that is expected to grow to 12.7 million by the year 2050 [38]. Alzheimer’s disease is a complex neurodegenerative disorder characterized histologically by extracellular amyloid beta (Aβ) plaques, intracellular tau tangles and neuroinflammation [39]. It has also been shown that cholesteryl esters accumulate within the brain of AD patients and mouse models [40,41,42]. Additionally, the e4 allele of apolipoprotein E (ApoE), the major extracellular cholesterol carrier in the brain, has been identified as the most prominent genetic risk factor for late onset AD [43,44]. The e4 allele leads to ApoE with structural and trafficking differences that ultimately lead to an altered intracellular lipid metabolism [45,46,47]. The exact cause of this metabolic disruption is not known, but each e4 allele present confers an increased AD risk [48]. The first attempts at targeting cholesterol metabolism in AD as a potential therapeutic were focused on treatment with statins as a means of reducing cellular cholesterol levels by block cholesterol synthesis. These studies show that the whole-cell reduction of cholesterol by extraction with cyclodextrin, or treatment with statins, reduce the processing of the amyloid precursor protein (APP) into toxic Aβ species [49,50,51,52]. Despite promising pre-clinical evidence that statins may reduce the risk of AD, studies in humans have been less convincing [53,54].

The connection between ACAT activity and AD was first demonstrated when it was observed that ACAT1/SOAT1 knockout cells had reduced Aβ production [55]. Since then, it has been shown that reducing ACAT1/SOAT1 activity by genetic manipulation or small molecule inhibition reduces amyloid pathology and rescues deficits seen in mouse models of AD [56,57,58,59]. Subsequent work has shown that ACAT1/SOAT1 inhibition leads to enhanced Aβ degradation by microglia and reduced tau burden in neurons as a result of enhanced autophagy and lysosomal biogenesis [60,61]. It was also shown that ACAT1/SOAT1 inhibition cleared cholesterol ester lipid droplets in microglia bearing mutant TREM2 [62], and in human IPSC-derived neurons from AD patients [63]. We understand that blocking ACAT1/SOAT1 activity in the context of AD proves beneficial; however, the initial molecular events linking ACAT1/SOAT1 inhibition and phenotypic rescue remain to be defined.

Here, we demonstrate that acute ACAT1/SOAT1 inhibition with small molecule inhibitors lead to the rapid accumulation of cholesterol as well as ACAT1/SOAT1 protein at MAM. This local cholesterol buildup also correlates with a strengthening in ER-mitochondria connectivity.

## 2. Results

### 2.1. Initial Observation of the ACAT1/SOAT1 Blockade Cholesterol Pool

ACAT1/SOAT1-specific small molecule inhibitors K-604 [64] and F12511 [65] are commonly used to study ACAT1/SOAT1 biology. Both compounds exhibit a preference for ACAT1/SOAT1 over ACAT2/SOAT2 and both have passed phase 1 clinical safety tests as anti-atherosclerosis therapeutics; reviewed in [66]. In order to measure ACAT activity independent of endogenous membrane lipid composition, a method has been developed that involves the solubilization of the ACAT1 membrane protein by CHAPS detergent and reconstitution into a mixed micelle solution with a defined lipid composition [67] (Appendix A). Here, we used N9 cells. These are an immortalized mouse microglia cell lines that share many characteristics with primary microglia [68], and have been used to study Aβ uptake and degradation in our lab previously [60]. Using the mixed micelle ACAT assay, we show that treatment with K-604 inhibits ACAT from N9 microglial cells with sub-micromolar efficacy (Figure 1A). When paired with an in-cell washout to remove K-604 prior to the reconstitution and measurement of enzyme activity, results show it takes about 60 min for the ACAT1/SOAT1 activity of K-604 treated cells to return to the level of uninhibited activity measured in vehicle-treated cells (Figure 1B). This demonstrates that K-604 can be removed from cells and ACAT by washing. 

Next, we wanted to monitor K-604 washout with a different assay by measuring ACAT activity in intact cells using a [^3^H] oleate pulse [69,70] (Appendix A). In this assay, cells are pulsed with [^3^H] oleate which is rapidly activated to [^3^H] oleate coenzyme A and subsequently used as a substrate for the production of cholesteryl esters, triglycerides and phospholipids. Surprisingly, upon drug washout, we observe that cells pre-treated with K-604 showed a robust spike in ACAT activity well above that of vehicle-treated cells (Figure 1C); this result was confirmed in SHSY5Y and CHO cells (Appendix A). The key difference between these two assays (Appendix A) is that the mixed micelle assay measures reconstituted ACAT activity with exogenous cholesterol provided in excess whereas the [^3^H] oleate pulse assay measures esterification of endogenous cholesterol from the membrane surrounding ACAT1/SOAT1. This suggests that upon inhibition, there is a buildup of cholesterol in the membrane surrounding ACAT1/SOAT1. When K-604 inhibition begins to be released by drug washout, the accumulated cholesterol becomes available as a substrate; this is observed as a spike in ACAT activity. This substrate buildup appears to occur rapidly as its presence is seen with K-604 treatments as short as 5 min (Appendix A). Similar experiments were performed with another ACAT1/SOAT1 inhibitor: F12511. The results showed that F12511 could not easily wash out of cells; a pre-treatment with this compound strongly inhibits ACAT activity after 8 h of repeated washing [71]. We attribute this to F12511′s higher binding affinity, and the fact that it is much more hydrophobic and cannot be removed from the membrane easily. 

We also wanted to understand where the cholesterol available to ACAT1/SOAT1 as a substrate comes from. To do this, we employed the use of small molecule inhibitors U18666A [72], and lovastatin [73]. At low concentrations, U18666A inhibits NPC1, a lysosomal protein responsible for the cellular distribution of lysosomal cholesterol [74,75]. Lovastatin inhibits HMG CoA reductase, the rate-limiting step in cholesterol synthesis [73]. Treatment with U18666A will effectively block the input of exogenously-derived cholesterol, while lovastatin treatment will effectively block the input of endogenously synthesized cholesterol. Here, we see that treatment with either inhibitor blocks ACAT activity, indicating that ACAT1/SOAT1 receives inputs from multiple cholesterol sources (Figure 1D).

### 2.2. Direct Observation of Cholesterol Buildup around ACAT1/SOAT1

In order to directly measure the substrate buildup around ACAT1/SOAT1, we used a fractionation approach combined with thin-layer chromatography (TLC) lipid analysis. First, using OptiPrep gradient ultracentrifugation [77], we were able to achieve the crude separation of cellular components and observed a significant increase in cholesterol in cell fraction #10, one of the fractions enriched in ACAT1/SOAT1 protein (Appendix A). Seeking a cleaner separation of cellular components, we took advantage of ACAT1/SOAT1′s presence on the mitochondria-associated ER membrane (MAM). The MAM fractionation technique is able to separate microsomes from MAM from mitochondria [17,78]. Indeed, we see that ACAT1/SOAT1 is enriched at the MAM along with other canonical MAM markers: sigma 1 receptor (Sigma1R) and acyl-CoA synthetase long-chain family member 4 (ACSL4) (Figure 2A,B). Interestingly, ACAT1/SOAT1 is selectively enriched at the MAM upon inhibition. Analyzing cholesterol in these fractions reveals a 20% increase in MAM cholesterol upon K-604 treatment (Figure 2C,D). Selective ACAT1/SOAT1 enrichment and cholesterol buildup at the MAM is confirmed with a second ACAT1/SOAT1 inhibitor F12511 (Figure 3). This is the first direct observation of cholesterol buildup around ACAT1/SOAT1 upon inhibition. Importantly, this cholesterol buildup at the MAM represents a redistribution of intracellular cholesterol as the whole cell cholesterol levels remained unchanged with acute (4 h) K-604 treatment (Figure 2E).

### 2.3. ACAT1/SOAT1 Inhibition Leads to Changes in ER-Mitochondria Connectivity

The close connection between the ER and mitochondria is a defining characteristic of the MAM. ER–mitochondria connectivity has been shown to be altered in Alzheimer’s models [3,79,80]. Additionally, we are starting to understand that there is a relationship between cellular cholesterol levels and ER-mitochondria connectivity [23,81]. Based on these findings, we wanted to understand whether or not ACAT1/SOAT1 inhibition would lead to changes in ER-mitochondria connectivity; something that would have important homeostatic implications. We performed mass spectrometry proteomic analyses of MAM fractions isolated from cells treated with K-604 or a vehicle. While we do not observe a significant change in any specific proteins using this technique, looking at the relative change of large functional groups of proteins stratified by a subcellular location reveals a notable increase in mitochondria proteins identified in this fraction when treated with K-604 (Figure 4A). This stands in contrast to the overall change in total proteins identified, as well as proteins identified from other subcellular compartments (Figure 4A). Changes in MAM proteins identified in this screen are shown in Appendix A. The overall increase in mitochondria proteins suggests a strengthening of the ER-mitochondria connection. This data was only suggestive though, so we next used a microscopic approach to investigate further. Using confocal fluorescent microscopy to look at the overlap between ER and mitochondria-localized markers as a measure of ER-mitochondria connectivity [82] in HMC3 cells (a human microglia cell line that is flat, making them better suited for imaging), we did see a trending increase in the overlap that does not quite reach statistical significance (*p* = 0.06) (Figure 4B,C). In order to definitively say whether or not there is a change in connectivity though, we turned to electron microscopy. Here, we analyzed the contact site at the closest point of contact between each mitochondria and the ER and, using an approach similar to that described in Lak et. Al. [81], we measured different contact site parameters (Figure 4D–H). We see that, with K-604 treatment, the overall percentage of mitochondria with a close contact (defined as a <30 nm intermembrane distance) increases compared to vehicle-treated cells (Figure 4E). Looking at other contact site characteristics, as previously measured [81], we see that ACAT1/SOAT1 inhibition has no effect on the length of the contact site (defined by a 30 nm intermembrane distance cutoff); however, when looking at the distribution of intermembrane distances measured within each contact site (between 0 and 100 nm), we see a multimodal distribution with notable differences between K-604- and vehicle-treated cells (Figure 4F,G). We stratified these distances into close (5–25 nm), intermediate (26–50 nm) and long-range (51–70 nm) distances, and were able to see significant increases in short and intermediate inter-membrane distances, while the number of long-range distances decreases (Figure 4G,H). This demonstrates that ACAT1/SOAT1 inhibition and resulting cholesterol accumulation at the MAM does lead to changes in ER-mitochondria connectivity.

## 3. Discussion

The link between cholesterol homeostasis and Alzheimer’s disease is well established, and yet, poorly understood. Attempts to change whole cell cholesterol levels by modulating synthesis with statins have shown initial promise at the pre-clinical stage, but benefits in the clinic remain elusive [83]. Targeting ACAT1/SOAT1 activity has been another promising avenue of therapeutic cholesterol modulation. Blocking ACAT1/SOAT1 activity by genetic ablation or pharmacological inhibition has been shown to reduce amyloid pathology in mice by lowering levels of the amyloid precursor protein (APP) and its toxic product, Aβ [56,57,59]. This could be a result of reduced APP processing [55] due to reduced APP palmitoylation and decreased localization to lipid rafts where it gets processed [84], and/or by altering plasma membrane cholesterol levels [85]. It could also be a result of the enhanced phagocytosis and subsequent degradation by upregulated autophagy and lysosomal biogenesis seen with ACAT1/SOAT1 inhibition in microglia [60]. This upregulated degradation machinery also leads to a reduction of intracellular tau in neurons [61]. The ACAT1/SOAT1 blockade has also been shown to enhance HMG-CoA reductase degradation by increasing levels of 24S-hydroxycholesterol [58] and suppress neuroinflammation by altering toll-like receptor 4 (TLR4) trafficking and activation [86]. Additionally, in a mouse model, genetic inactivation of ACAT1/SOAT1 was recently shown to benefit a rare pediatric neurodegenerative disease, Niemann-Pick type C1 (NPC1) [87]. The effects of ACAT1/SOAT1 inhibition are undoubtedly beneficial in disease contexts, but it is difficult to tie these results together with a single common thread. Studies have mostly focused on the end-result of phenotypic rescue without understanding the molecular explanation for these results.

Here, we provide important evidence demonstrating the initial molecular and cellular responses to ACAT1/SOAT1 inhibition. Firstly, we observed cholesterol accumulation in the metabolically important subdomain of the ER known as the mitochondria-associated ER membrane (MAM). Changing the amount of cholesterol in a membrane will alter that membranes’ physical properties. Cholesterol creates order in the membrane. This will have an effect on membrane thickness and phase, which governs fluidity and flexibility, and will also change the membrane permeability of hydrophilic molecules such as water and molecular oxygen, thus altering hydrophobicity within the membrane [3,88]. Additionally, cholesterol interacts with the transmembrane domains of membrane-embedded proteins in an annular and non-annular fashion by general electrostatic interactions or with conserved domains [89,90,91]. These membrane characteristics will have an effect on protein localization and function [4,12,92,93,94,95].

An example of cholesterol-driven protein localization is shown here with the selective enrichment of ACAT1/SOAT1 in the MAM fraction upon cholesterol accumulation. ACAT1/SOAT1′s role in the cell can be boiled down to its ability to keep ER–cholesterol levels low by removing it from the membrane. We know ACAT1/SOAT1 has at least two cholesterol binding sites: a substrate binding site and an allosteric site [28,96]. It is likely that the cholesterol-associated enrichment of ACAT1/SOAT1 in the MAM is driven by ACAT1/SOAT1′s preference for the physical characteristics of membranes with higher cholesterol content. This could potentially explain, at least in part, some of the reports shown in the literature. For example, it has been previously observed that MAMs are altered in Alzheimer’s disease [97], and this correlates with higher ACAT1/SOAT1 activity [37,79] and cholesterol accumulation within the MAM [98], similar to what is observed here. Exactly what drives this cholesterol accumulation and the degree to which ACAT1/SOAT1 activity is a symptom or driver of the disease remains unclear. It is possible that, in the diseased state, APP or its 99-aa C-terminal fragment (C99) product [99] leads to ATAD3A clustering which alters cholesterol turnover mediated by the ER–mitochondria contact site [23]. Blocking this cholesterol disposal pathway causes cholesterol accumulation at the MAM which leads to ACAT1/SOAT1 enrichment and an observed increase in ACAT1/SOAT1 activity resulting in higher cholesteryl ester levels. 

The mechanisms for cholesterol-mediated strengthening of the MAM–mitochondria interaction are not known. It is possible that increased cholesterol will alter clustering, recruitment or stabilization of the protein tethers that bridge the two organelles, but the exact players involved remain to be seen. Additionally, the similarities between harmful MAM disruptions observed in AD (i.e., increased ER–mitochondria contact, cholesterol buildup, enhanced calcium transfer and phospholipid synthesis [23,37,100]) and the beneficial response to ACAT1/SOAT1 inhibition (i.e., increased ER–mitochondria contact and cholesterol buildup) are at odds with each other and require further investigation in cells modeling the disease. It is possible that proximity-facilitated processes, such as calcium and phospholipid transfer, become enhanced in the disease state while other MAM functions and regulation become disrupted by an over-accumulation of APP and its C99 product [99]. ACAT1/SOAT1 inhibition could help to alleviate this by providing an additional functional MAM domain at which these processes can resume while at the same time reducing MAM levels of APP and subsequent processing. This brings up the need for a more in-depth functional analysis of a “healthy” vs “unhealthy” strengthening of the MAM in healthy and disease contexts in future studies.

It is vital that we exhaust all therapeutic avenues when it comes to finding a treatment for Alzheimer’s disease. Not only will this be important to the millions of patients and their families affected by this devastating disease, but this work could also benefit patients who suffer from other complex neurodegenerative disorders. Despite the presence of different protein pathologies that affect different regions of the brain, neurodegenerative disorders such as Alzheimer’s disease, Parkinson’s disease (PD) and amyotrophic lateral sclerosis (ALS) have shockingly similar cellular and molecular deficits that can be linked to disruptions in the MAM domain and ER–mitochondria connectivity [97]. Beneficial modulation of the MAM domain could translate to these other cureless diseases.

## 4. Materials and Methods

### 4.1. Materials

Antibodies: The rabbit polyclonal antibody (DM102) against the N-terminal fragment (1–131) of human ACAT1 was described previously [29]. FACL4/ACSL4 (155282) is from Abcam. Sigma1R (SC-137075), Na/K ATPase (SC-21712), Cytochrome C oxidase (SC-58347) and Tom20 (SC-17764) are from Santa Cruz (Dallas, TX, USA). Syntaxin 6 is a kind gift from Andrew Paden’s Lab. Vinculin (05-386) is from Millipore (Burlington, MA, USA). Secondary Antibodies Goat Anti-Mouse IRDye 680RD (926-68070) and Goat Anti-Rabbit IRDye 800CW (926-32211) are from Li-Cor Biosciences (Lincoln, NE, USA). Other materials: oleic acid, coenzyme A trilithium salt, sodium taurocholate, triglyceride, cholesterol, cholesteryl oleate, fatty acid-free bovine serum albumin, CHAPS detergent, Percoll and the protease inhibitor cocktail were from MilliporeSigma (Burlington, MA, USA). OptiPrep was from Cosmo Bio USA (Carlsbad, CA, USA). Fugene 4K was from Promega (Madison, WI, USA). [^3^H] Oleic acid is from Perkin Elmer (Waltham, MA, USA). The [^3^H] oleoyl-CoA was synthesized as described previously [101]. The silica thin layer chromatography plates are from Analtech (Newark, DE, USA). Mito-BFP (described in [102]; addgene #49151), containing TagBFP localized to the mitochondria matrix with 1–22 amino acid sequence from *S. cerevisiae* COX4, was a gift from Henry Higgs. KDEL-RFP containing TagRFP localized to the ER by KDEL ER retrieval motif was a gift from Henry Higgs. K-604 and F12511 were custom synthesized by WuXi AppTec (Shanghai, China); based on HPLC/MS and NMR profiles, purity of F12511 was 98% and in stereospecificity; purity of K-604 was 98%.

### 4.2. Cell Culture

All cell lines were maintained at 37 °C under humidified conditions and 5% CO2. The N9 cells were maintained in RPMI (Corning, Corning, NY, USA) with 10% calf serum (Atlanta Biologicals, Flowery Branch, GA, USA). The HMC3 cells were maintained in DMEM medium (Corning, Corning, NY, USA) supplemented with 10% calf serum. CHO cells were maintained in DMEM:F-12 medium (Corning, Corning, NY, USA) 1:1 with 10% calf serum. SHSY5Y cells were maintained in DMEM:F12 (Corning, Corning, NY, USA) 1:1 with 10% calf serum and non essential amino acids (Thermo Fisher Scientific, Waltham, MA, USA).

### 4.3. Western Blot

Cells were grown, treated and collected by scraping into PBS. Cells were pelleted and resuspended in 10% SDS. The protein concentration was determined using a modified Lowry assay and equal amounts of protein were aliquoted and prepared for SDS page analysis by adding DTT and SDS loading dye. After running, the proteins were transferred to nitrocellulose membrane (Cytiva, Marlborough, MA, USA). After transfer, the membrane was dried, blocked with 5% milk in TBST and incubated with a primary antibody for 3 h to overnight. After the wash, the membrane was incubated with Li-Cor fluorescent secondary antibodies, washed again and visualized on Li-Cor Odyssey CLx.

### 4.4. Measuring ACAT Activity by Mixed Micelle Assay

The mixed micelle assay was performed as described [67]. Briefly, cells were grown in 6-well plates at least 48 h before experimenting. At t = −24 h fresh media was added, and cells were treated with small molecule inhibitors as described in the figures. Cells were lysed in 75 µL buffer containing 50 mm tris, 1 mM EDTA, 1 M KCl, 2.5% CHAPS, with protease inhibitor. Aliquots were removed for protein determination by modified Lowry assay. On ice, 10 µL of lysate was added to 100 µL of prepared mixed micelle solution containing 11.1 mM phosphatidylcholine, 1.8 mM cholesterol and 9.3 mm taurocholate. The samples were then vortexed and equilibrated for 5 min on ice. An amount of 10 nm [^3^H] oleoyl-coenzymeA/BSA was added, vortexed and placed in a 37 °C shaking water bath for exactly 10 min before stopping the reaction with chloroform:methanol (2:1). To each sample, 50 µg cold cholesteryl oleate (carrier) was added, followed by a chloroform: methanol lipid extraction. Isolated lipids were spotted on TLC plates and ran using a petroleum ether: ethyl ether: acetic acid (90:10:1) solvent system. Lipids were visualized with iodine stain, the cholesterol ester band was scraped, and [^3^H] was measured with a scintillation counter.

### 4.5. Measuring ACAT Activity by [^3^H] Oleate Pulse Assay

The [^3^H] oleate pulse assay was performed as described [69]. Briefly, cells were plated at least 48 h before lyse. Cells were grown in 12-well dishes (N9, CHO cells) or 6-well dishes (SHSY5Y, HMC3 cells) until 90% confluent. Cells with wash had the drug-containing media replaced by a conditioned drug-free media for given amounts of time. Cells were pulsed with 10 μL of 10 mM [^3^H] oleate/BSA (containing about 4 μCi) for 30 min (N9, CHO cells) or 60 min (SHSY5Y, HMC3cells) in a 37 °C water bath with 5% CO_2_. The reaction was stopped by placing cells on ice, washed 3 times with cold PBS, and lysed with 0.2 M NaOH. An aliquot was taken for protein determination with modified Lowry assay, and cold lipid carriers were added for each species to be analyzed followed by chloroform: methanol (2:1) extraction and spotted on TLC plates. Neutral lipids were separated using the solvent system; petroleum ether: ethyl ether: acetic acid (90:10:1). Lipids were visualized with an iodine stain. Bands were marked, iodine was evaporated off and bands were scraped and [^3^H] was measured with a scintillation counter.

### 4.6. Optiprep Fractionation

Cells were isolated on ice by washing 2× with cold PBS and 1× with homogenization buffer (HB; 20 mM tris pH 7.4, 250 mM sucrose, 1 mM EDTA, protease inhibitor as needed). Cells were collected in HB + protease inhibitor and homogenized in Dounce homogenizer. A post-nuclear supernatant was carefully layered on top of a prepared 5%, 10%, 15%, 20%, 25% (top to bottom) OptiPrep gradient in a clear ultracentrifuge tube. Samples were spun at 40,000 rpm in a Beckman L8-M Ultracentrifuge for 3 h at 4 °C under vacuum in a SW41 swing rotor. Fractions were collected by aliquoting 800 μL from the top of the ultracentrifuge tube. Aliquots of each fraction were removed for protein determination with a modified Lowry assay. The samples were analyzed for protein by Western blot. For cholesterol analysis, the lipids were extracted with chloroform: methanol (2:1) and spotted on a TLC plate. Cholesterol was separated using the solvent system hexanes: ethyl ether: acetic acid (30:20:1). To visualize the lipids, plates were submerged briefly with 3% Copper acetate (*w*/*v*) in 8% phosphoric acid (*v/v*) and heated at 180 °C for approximately 15 min to char [103].

### 4.7. MAM Fractionation

MAM fractionation was performed as described [92]. Briefly, cells were washed 2× with cold PBS on ice, collected by scraping in homogenization buffer (HB; 250 mM sucrose, 10 mM HEPES pH 7.4) and lysed with Dounce homogenizer. The nuclei and whole cells were removed with 2 × 600× *g* spins at 4 °C. Supernatant was moved to ultracentrifuge tube. Using Beckman L8-M Ultracentrifuge with 70.1 Ti rotor, crude mitochondria were pelleted by spinning at 12,000 RPM for 20 min at 4 °C under a vacuum. Supernatant was removed and plasma membrane and lysosomes were removed by spin at 14,800 RPM for 30 min at 4 °C under a vacuum. Supernatant was removed and microsome fraction was pelleted by spinning at 39,000 RPM for 1 h at 4 °C under vacuum. The crude mitochondria were resuspended in HB and pelleted again to remove contaminants by spinning at 12,000 RPM for 20 min at 4 °C under vacuum. Crude mitochondria were resuspended in isolation buffer (250 mm mannitol, 5 mm HEPES pH 7.4, 0.5 mm EGTA pH 7.4) by gentle vortexing and layered on top of a percoll solution (225 mm mannitol, 25 mm HEPES pH 7.4, 1 mm EGTA pH 7.5, 30% percoll). The MAM and mitochondria were separated with a SW41 swing bucket rotor by spinning at 38,000 RPM for 30 min at 4 °C under vacuum with acceleration and deceleration set to 1. MAM band is seen 2/3 of the way towards the top, mitochondria are at the bottom. Fractions were washed in PBS 3× by spin at 10,300× *g* for 20 min before final pelleting. The protein concentration was determined using a modified Lowry assay. The protein was analyzed via Western blot. For cholesterol analysis, lipids were extracted with chloroform: methanol (2:1) and spotted on a TLC plate. Cholesterol was separated using the solvent system hexanes: ethyl ether: acetic acid (30:20:1), and plates were dried. To visualize lipids, plates were submerged briefly in with 3% Copper acetate (*w/v*) in 8% phosphoric acid (*v*/*v*) and heated at 180 °C for approximately 15 min to char [103].

### 4.8. Proteomic Analysis of MAM Fractions

MAM fractions were isolated and ACAT1 enrichment was confirmed using Western blot. An amount of 100 µg of protein/sample were dried with a speedvac and sent to the IDeA National Resource for Quantitative Proteomics for analysis. Protein samples were reduced, alkylated, and purified by chloroform/methanol extraction prior to digestion with sequencing grade trypsin (Promega). The resulting peptides were labeled using a tandem mass tag 10-plex isobaric label reagent set (Thermo Fisher Scientific, Waltham, MA, USA) and combined into one multiplex sample group. The labeled peptides were separated into 46 fractions on a 100 × 1.0 mm Acquity BEH C18 column (Waters) using an UltiMate 3000 UHPLC system (Thermo Fisher Scientific, Waltham, MA, USA) with a 50 min gradient from 99:1 to 60:40 buffer A:B ratio under basic pH conditions, then consolidated into 18 super-fractions. Each super-fraction was then further separated by reverse phase XSelect CSH C18 2.5 um resin (Waters) on an in-line 150 × 0.075 mm column using an UltiMate 3000 RSLCnano system (Thermo Fisher Scientific, Waltham, MA, USA). Peptides were eluted using a 75 min gradient from 98:2 to 60:40 buffer A:B ratio. Eluted peptides were ionized by electrospray (2.4 kV) followed by mass spectrometric analysis on an Orbitrap Eclipse Tribrid mass spectrometer (Thermo Fisher Scientific, Waltham, MA, USA) using multi-notch MS3 parameters. MS data were acquired using the FTMS analyzer in top-speed profile mode at a resolution of 120,000 over a range of 375 to 1500 m/z. Following CID activation with normalized collision energy of 31.0, MS/MS data were acquired using the ion trap analyzer in centroid mode and normal mass range. Using synchronous precursor selection, up to 10 MS/MS precursors were selected for HCD activation with a normalized collision energy of 55.0, followed by acquisition of MS3 reporter ion data using the FTMS analyzer in profile mode at a resolution of 50,000 over a range of 100–500 m/z.
Buffer A = 0.1% formic acid, 0.5% acetonitrile
Buffer B = 0.1% formic acid, 99.9% acetonitrile

#### Both Buffers Adjusted to pH 10 with Ammonium Hydroxide for Offline Separation

Proteins were identified and MS3 reporter ions quantified using MaxQuant (Max Planck Institute) against the UniprotKB Mus musculus, January 2022) database with a parent ion tolerance of 3 ppm, a fragment ion tolerance of 0.5 Da and a reporter ion tolerance of 0.003 Da. Scaffold Q + S (Proteome Software) was used to verify MS/MS-based peptide and protein identifications (protein identifications were accepted if they could be established with less than 1.0% false discovery and contained at least 2 identified peptides; protein probabilities were assigned by the Protein Prophet algorithm [104] and to perform reporter ion-based statistical analysis.

Protein TMT MS3 reporter ion intensity values are assessed for quality and normalized using ProteiNorm [105]. The data was normalized using cyclic loess [106] and statistical analysis was performed using Linear Models for Microarray Data (limma) with empirical Bayes (eBayes) smoothing to the standard errors [106]. Proteins with an FDR adjusted *p*-value < 0.05 and a fold change > 2 are considered to be significant.

Proteins were annotated with g:Profiler web browser-based software using g:GOSt functional profiling [107]. Groups of proteins were identified using GO term IDs as follows: nucleus, GO:0005634; mitochondrion, GO:0005739; lysosome, GO:0005764; endosome, GO:0005768; autophagosome, GO:0005776; peroxisome, GO:0005777; endoplasmic reticulum, GO:0005783; Golgi apparatus, GO:0005794; lipid droplet, GO:0005811; cytosol, GO:0005829; cytoskeleton, GO:0005856.

### 4.9. Confocal Microscopy

HMC3 cells were seeded in a 6-well dish and transfected with plasmid DNA using Fugene 4K reagent in serum-free Opti-MEM (Thermo Fisher Scientific, Waltham, MA, USA) for 4 h before replacing with full media. After 24–48 h, cells were trypsinized and re-plated onto poly-D-lysine coated coverslips and grown for at least another 24 h. Samples were prepared for imaging as previously described [82]. Briefly, after drug treatments, cells were washed with PBS and fixed with 1% glutaraldehyde in PBS for 10 min and treated with 1 mg/mL sodium borohydride in PBS 3 × 15 min to quench autofluorescence. Images were taken using Andor (Belfast, Northern Ireland) spinning disk on a Nikon (Melville, NY, USA) Eclipse Ti base equipped with Andor Zyla sCMOS camera and Yokogawa (Tokyo, Japan) Power supply using 60 × 1.4 NA Plan-Apo γ Nikon oil objective.

ER–mitochondria overlap analysis was performed using ImageJ image analysis software (version: 2.9.0/1.53t). Individual mitochondria were isolated from the cell periphery and images were batch-processed using a macro written to auto-threshold and create binary masks for ER and mitochondria signals followed by a calculation of the area of colocalized pixels. This area was standardized back to the mitochondria pixel area to measure the % mitochondria overlapping with ER.

### 4.10. Electron Microscopy

A431 cells were grown on fibronectin-coated glass-bottom Mattek (Ashland, MA, USA) dishes, treated with conditions as described, and fixed with freshly prepared fixative: 2.5% glutaraldehyde, 3.2% paraformaldehyde in 0.1 M sodium cacodylate pH 7. Cells were postfixed for 1 h on ice with 1% OsO4 in 0.1 M sodium cacodylate pH 7.2, rinsed 2 × 5 min at room temp in 0.1 M sodium cacodylate pH 7.2, rinsed 2 × 1 min in water, incubated overnight at RT in 2% aqueous uranyl acetate, dehydrated in graded ethanol (50%, 70%, 95%, 2 × 100%) and embedded in LX-112 resin using Thompson molds. Sections parallel to the cellular monolayer were obtained using a Leica Ultracut 7 ultramicrotome, mounted on carbon–coated 200-mesh copper grids (Ted Pella), stained with a mixture of rare earths (UranylLess—EMS) and lead citrate (EMS), and examined and imaged under a Helios 5CX electron microscope using a STEM3+ detector.

The images analyzed were obtained from duplicate biological replicates. Electron micrograph analysis was performed using ImageJ image analysis software (version: 2.9.0/1.53t). Individual mitochondria were analyzed by identifying the closest ER structure, and measuring distances flanking either side of the minimum, inspired by [81]. With the use of an ImageJ macro, the mitochondria and ER membrane surfaces were identified by hand. Every 2 pixels (7.73 nm) along the mitochondria surface, a line representing the intermembrane distance was drawn to the closest point on the ER (Figure 4D). The contact site length was calculated on the mitochondria surface by measuring the number of consecutive intermembrane distances under 30 nm.

### 4.11. Data Analysis and Visualization

Data analysis and statistical comparisons were made using a two-tailed, unpaired Student’s t-test performed with Microsoft Excel or GraphPad Prism 9. All graphs and data visualization were completed with GraphPad Prism 9.

## 5. Conclusions

The benefits of ACAT1/SOAT1 inhibition in diseases have been known for some time, but until now the immediate downstream molecular and cellular responses leading to these benefits were unknown. This study provides the first known example of precise modulation of cholesterol at the mitochondria-associated ER membrane (MAM) and shows cholesterol accumulation at the MAM and a strengthening of the ER-mitochondria contact site are some of the first downstream events following ACAT1/SOAT1 inhibition. Based on genome-wide association (GWAS) studies, several genes identified as risk factors for Alzheimer’s disease are closely linked with cholesterol/lipid homeostasis, and yet, the effects of total cholesterol reduction therapy by statins have shown mixed results. Our current study provides a strong rationale that we should perhaps be looking at altering localized cholesterol within cells.

## Figures and Tables

**Figure 1 ijms-24-05525-f001:**
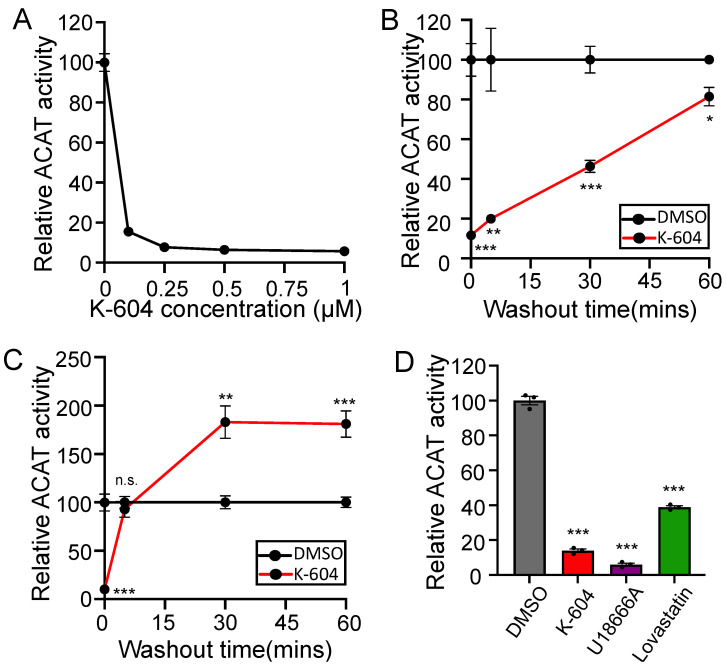
Initial observation and characterization of the ACAT1 blockade cholesterol pool. (**A**) N9 cells were treated with various concentrations of K-604 or DMSO vehicle control for 8 h prior to lysis followed by reconstitution and measurement of ACAT activity by mixed micelle ACAT assay. *n* = 8 (**B**) N9 cells were treated with 0.5 µM K-604 or DMSO vehicle control for 4 h prior to aspiration and incubation with drug-free media for given amounts of time. Cells were then lysed followed by reconstitution and measurement of ACAT activity by mixed micelle ACAT assay. *n* = 3 (**C**) N9 cells were treated with 0.5 µM K-604 or DMSO vehicle control for 4 h prior to aspiration and incubation with drug-free media for given amounts of time. ACAT activity was measured in intact cells by [^3^H] oleate pulse. *n* = 6 (**D**) N9 cells were treated with DMSO, 0.5 µM K-604, 70 nM U18666A or 50 µM lovastatin and 230 µM mevalonate for 4, 4, 8 and 24 h respectively. Mevalonate is added during lovastatin treatment to allow for the synthesis of non-cholesterol products downstream of HMGCR [76]. ACAT activity was measured in intact cells by [^3^H] oleate pulse. *n* = 3. Error bars represent SEM. Note for (**A**–**C**): all datapoints have error bars, some are smaller than the data point and are difficult to see. *p*-value determined using Student’s *t*-test; n.s.= not significant; * *p* < 0.05; ** *p* < 0.01; *** *p* < 0.001.

**Figure 2 ijms-24-05525-f002:**
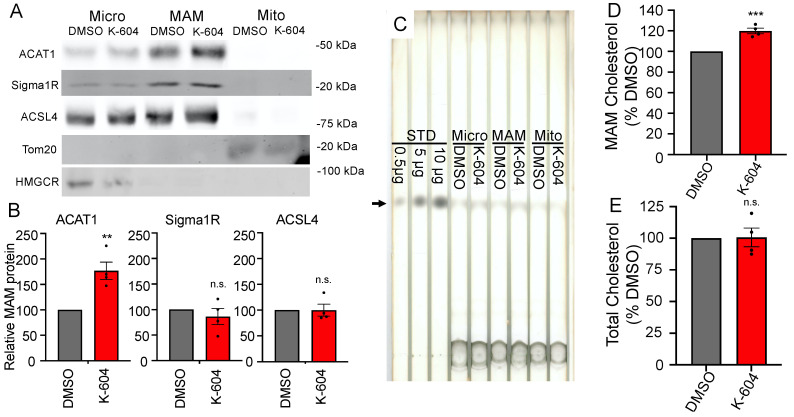
Direct observation of cholesterol accumulation at the MAM with K-604. N9 cells were treated with 0.5 µM K604 or DMSO vehicle control for 4 h. (**A**–**D**) After drug treatments, cells were lysed and subject to MAM fractionation protocol to isolate microsomal, MAM and mitochondria fractions: (**A**) Representative Western blots depicting distribution between fractions of MAM proteins ACAT1, Sigma1R and ACSL4, mitochondria protein Tom20 and ER protein HMGCR, 200 µg protein per lane. (**B**) Quantification of Western blots for ACAT1, Sigma1R and ACSL4 in the MAM fraction. For each experiment, the protein band for K-604-treated cells was standardized to DMSO-treated cells *n* = 4. (**C**) Representative TLC plate. Arrow denotes cholesterol band. RF = 0.3, 250 µg protein per lane. (**D**) Quantification for bulk cholesterol measured in the MAM fraction. *n* = 4. (**E**) After drug treatment, cells were lysed and subject to cholesterol analysis without fractionation. *n* = 3. Error bars represent SEM. *p*-value determined using Student’s *t*-test; n.s.= not significant; * *p* < 0.05; ** *p* < 0.01; *** *p* < 0.001.

**Figure 3 ijms-24-05525-f003:**
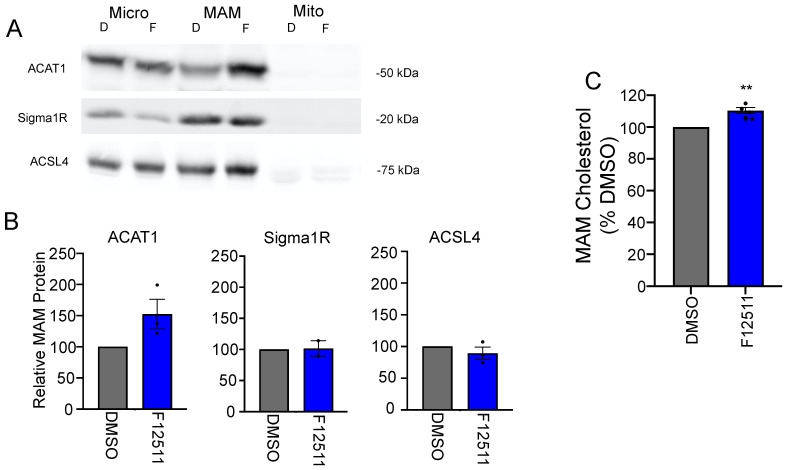
Direct observation of cholesterol accumulation at the MAM with F12511. N9 cells were treated with 0.5 μm F12511 or DMSO vehicle control for 4 h. (**A**–**C**) After drug treatments, cells were lysed and subject to MAM fractionation protocol to isolate microsomal, MAM, and mitochondria fractions. (**A**) Representative Western blots depicting ACAT1, Sigma1R and ACSL4 distribution within fractions, 200 µg protein per lane (**B**) Quantification of Western blots for ACAT1, Sigma1R and ACSL4 in the MAM fraction. For each experiment, the protein band for K-604-treated cells was standardized to DMSO-treated cells *n* = 3. (**C**) Quantification for bulk cholesterol measured in the MAM fraction. *n* = 4. Error bars represent SEM. *p*-value determined using Student’s *t*-test; * *p* < 0.05; ** *p* < 0.01.

**Figure 4 ijms-24-05525-f004:**
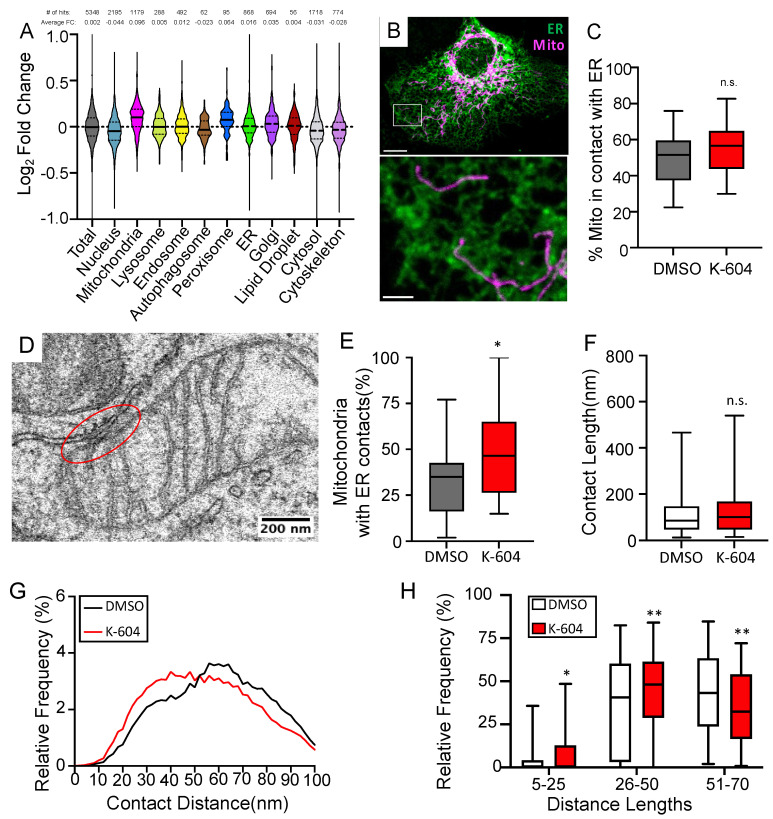
ACAT1 inhibition alters ER–mitochondria connectivity. (**A**) Proteomic analysis of MAM fractions of K-604 or DMSO-treated N9 cells. GO:cellular component annotations were assigned with g:GOSt functional enrichment program on the g:Profiler website. Violin plot depicts distribution of Log2Fold Change between DMSO- and K-604-treated cells with black bar representing mean for proteins stratified based on subcellular location. Number of proteins identified and average fold change listed above. *n* = 5. (**B**) Representative confocal image of fixed HMC3 (human microglia cell line) cells expressing KDEL-RFP and mitochondria-localized BFP representing ER in green, and mitochondria in magenta, respectively. Whole cell (above) scale bar = 10 µm. Inset (below) scale bar = 2 µm. (**C**) Quantification of overlap between ER and mitochondria in the cell periphery as the area of overlapping ER and mitochondria signal standardized to the total mitochondria area. DMSO, *n* = 60; K-604, *n* = 60. (**D**) Representative electron micrograph of A431 cell with mitochondria pseudocolored in magenta, ER pseudocolored in green and example contact site circled in red (**E**) Quantification of the percentage of mitochondria with close contacts as defined by 30 nm cutoff per field of view. DMSO, *n* = 166 mitochondria analyzed in 22 fields of view; K-604, *n* = 155 mitochondria analyzed in 24 fields of view. (**F**) Quantification of the contact site length. Continuous length along the membrane was measured where the ER and mitochondria were less than 30 nm apart. DMSO, *n* = 72 contact sites; K-604, *n* = 88 contact sites. (**G**) Histogram showing the distribution of all contact site distances measured between 0 and 100 nm. Bold trace represents running average (window = 7). DMSO, *n* = 217 contact sites analyzed; K-604, *n* = 194 contact sites analyzed. (**H**) Quantification of contact site distance distribution stratified into close (5–25 nm), intermediate (26–50 nm) and long-range (51–70 nm) categories. DMSO, *n* = 174 contact sites analyzed; K-604, *n* = 175 contact sites analyzed. Error bars represent SEM. *p*-value determined using Student’s *t*-test; n.s.= not significant; * *p* < 0.05; ** *p* < 0.01.

## Data Availability

Proteomics data generated in this paper can be found in the MassIVE database, Identifier number: MSV000091335.

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
