# Peer review of "Acute ACAT1/SOAT1 Blockade Increases MAM Cholesterol and Strengthens ER-Mitochondria Connectivity"

_ijms, 2023, doi:10.3390/ijms24065525_

Round 1
Reviewer 1 Report
In this study, Harned et al. show that modulating cholesterol levels at the mitochondria-associated ER membrane (MAM) influences the ER-mitochondria connection. Specifically, they found that selective inhibition of ACAT1 increases local cholesterol concentration at the MAM. This increased local cholesterol increases the number of ER-mitochondria contact sites and shortens the distance between the two organelles, strengthening the ER-mitochondria connectivity. The study proposes a potentially therapeutic role for ACAT1 inhibition and the pathogenesis of Alzheimer’s disease.
There is clear role of dysregulated cholesterol metabolism in Alzheimer’s disease and yet the effects of modulating total cholesterol in therapies has not yielded clear improvements. This study provides strong evidence that we should perhaps be looking at localized cholesterol instead. I think this study will of great interest to field. The manuscript is well written, and the data supports the main conclusions. However, a few issues should be addressed to make this manuscript suitable for publication. Below is a list of comments that I hope the authors will find helpful.
Major comments:
1. Figure 1A; There are error bars for the control (0 uM K-604) but not for the rest of the experiment. The authors should revise the figure to include the error bars for all concentrations measured,
2. For Figure 1D, cells are treated with U18666A or lovastatin to block the input of exogenously or endogenously synthesized cholesterol. However, it’s not clear why the cells are also treated with mevalonate (line 158). I recommend the authors provide a more in-depth explanation (eg. HMG-CoA Reductase converts HMG-CoA into mevalonate for cholesterol synthesis).
3. In figure 2A, the authors show striking enrichment of the ACAT1 on the MAM that goes up with K-604 treatment. But in supplemental figure S3A, using a different fractionation protocol, ACAT1 is enriched on fractions 8/9 which the authors designate as ER (MAM being fractions 10/11) and do not see a differences in ACAT1 (levels or enrichment in different fractions). Can the authors explain this discrepancy?
4. Along the same lines, for Figures 2A and 3A, can the authors include a negative control that is present in the ER but absent from the MAM fraction? This would strengthen the confidence in the fractionations (if such a control exists).
Minor comments:
1. In line 144 the word ‘with’ can be removed.
2. In the results for the ACAT activity with F12511, the wrong figure is referenced (lines 145-147).
3. In several figures the effect sizes of the quantification appear smaller than they are because the Y-axis goes much higher than the last datapoint. For example, Figure 3C, a quick glance looks like no difference. I highly recommend the authors revise many of their graphs to use the full range of the Y-axis.
4. The authors use mitochondrial-localized BFP to measure ER-mito contact sites by confocal. In the methods, they reference where they obtained the plasmid (Henry Higgs ref 96). The authors need to include more information on the plasmid. What is the protein or targeting motif used to get the BFP to the mitochondria?
5. Figure 4D, it would be nice to include this image without the membranes/contact sites marked as well. Otherwise, you can’t see the membranes being quantified. This could go in the main figure or supplemental.
Reviewer 2 Report
Review report Harned et al.
Paper presents an interesting data on cholesterol intermembrane turnover and its role in intracellular structures interactions. The rationale resulting from introduction is clear. However, suggested link to AD-like mechanisms is not justified by the multiple cell lines model used. The aim for using 4-5 different cell lines must be clarified either in results or/and in discussion.
The use of several cell lines should be clearly explained on the basis of technical problems?, specific properties of each are or something else?
Elegant experiments with different specific inhibitors of ACAT/SOAT, demonstrate well the existence of multiple mechanisms contributing to its activity. Involvement of pro--inflammatory microglia in AD pathology is well known. However, shifts in ACAT/SOAT levels explaining mechanism of N9 viability do not justify extensive discussion of AD mechanisms.
Both introduction and the discussion seem to be out of focus of conducted experiments
Obligatory changes should not expand the article volume. Paper is feasible for publication after careful revision.
Reviewer 3 Report
My notes are attached here. Thanks. ReviewerXX.
